# Revisiting the Food- and Nutrition-Related Curriculum in Healthcare Education: An Example for Pharmacy Education

**DOI:** 10.3390/pharmacy9020104

**Published:** 2021-05-22

**Authors:** Eline Tommelein, Marthe De Boevre, Lize Vanhie, Inge Van Tongelen, Koen Boussery, Sarah De Saeger

**Affiliations:** 1Centre of Excellence in Mycotoxicology & Public Health, Faculty of Pharmaceutical Sciences, Ghent University, Ottergemsesteenweg 460, B-9000 Ghent, Belgium; eline.tommelein@vub.be (E.T.); Lize.vanhie@ugent.be (L.V.); Sarah.desaeger@ugent.be (S.D.S.); 2Laboratory for Pharmaceutical Chemistry, Drug Analysis and Drug Information (FASC), Department of Pharmaceutical Sciences (FARM), Faculty of Medicine and Pharmacy, Vrije Universiteit Brussel, Laarbeeklaan 103, 1090 Jette, Belgium; 3Pharmaceutical Care Unit, Faculty of Pharmaceutical Sciences, Ghent University, Ottergemsesteenweg 460, B-9000 Ghent, Belgium; Inge.vantongelen@ugent.be (I.V.T.); Koen.boussery@ugent.be (K.B.)

**Keywords:** food science, pharmacy education, community pharmacy, nutrition, pharmaceutical care

## Abstract

**Objective**: This study aimed to obtain an objective overview of nutritional topics discussed in community pharmacies to adapt the nutrition-related course content in pharmacy education. **Methods**: We performed an observational study between July 2014 and April 2015 in 136 community pharmacies in Belgium. During four months, each pharmacy intern recorded the first two food- and nutrition-related cases with which they were confronted. Each case was classified into one of 18 categories. **Results**: 1004 cases were included by 135 pharmacy interns. The most often discussed subjects include “food supplements” (38%), “baby food” (19%), and “healthy food and nutritional recommendations” (11%). In 45% (447/1004) of all cases, pharmacy interns were able to immediately discuss the cases without searching for additional information. Eventually, after looking up extra information, 95% (958/1004) of cases could be answered. **Conclusions**: Food- and nutrition-related cases are discussed in primary healthcare. We recommend food- and nutrition-related courses in the curriculum of every healthcare profession.

## 1. Introduction

Nowadays, consumers are exposed to controversial information on healthy food and nutrition through popular media [1]. The application of social media in various domains is an emerging trend due to a massive volume of available data, accessibility, and interaction [2,3]. This high volume of media coverage, however, has not brought clarity to or improved understanding of “healthy food and nutrition”, while it is a crucial factor for consumers’ behavior and health [4,5]. The Western diet, with high intakes of meat, fat, and sugar, is a risk for individual health and social systems [6,7]. A balanced diet, along with proper physical activity and psycho-emotional well-being is key to overall well-being [6,7]. It remains challenging to effectively communicate “nutritional health” that serves public understanding. Only an integrated and interdisciplinary approach in nutritional health communication will result in powerful and sustainable effects on the public’s understanding, on its behavior and consequent application and, ultimately, on its well-being [8,9].

Besides dieticians, patients turn to different healthcare providers for food- and nutrition-related advice, such as physicians, nurses, midwives, or pharmacists. As advisors, it is implicative that healthcare workers take responsibility in providing patients with evidence-based answers and recommendations concerning food- and nutrition-related topics. Studies show that, for example, pharmacists often feel uncertain regarding nutritional advice [10,11]. Therefore, to begin with, health education should be well aligned with nutritional topics addressed in daily practice. To the best of our knowledge, no research has been performed to identify the nutritional topics with which healthcare workers most often come into contact. This makes it utterly difficult to have an insight into what advice is expected of first-line healthcare providers about nutritional topics, as well as which topics should be included in the curricula.

In this research, we focused on the community pharmacist as an example of a healthcare worker in primary care. In the last few decades, the pharmacist’s role has changed from the preparation, supply, and control of drugs to functioning as a “pharmacotherapy manager” [12]. The pharmacist needs to assure that drug therapy is appropriate, effective, safe, and convenient for the patient, thus improving medication outcome, as well as quality of life. In addition, the pharmacist acts as a health promoter through their accessible advice. This patient-centered approach is called “pharmaceutical care” [12].

The aim of this observational study is to obtain an objective overview of all nutritional topics discussed in community pharmacies as an example of a primary healthcare point. In addition, this study aimed to verify the current knowledge of pharmacy interns on nutritional topics in the setting of a community pharmacy, and their ability to discuss nutritional cases. In addition, the students sought for additional information to elaborately discuss a patient case. The outcomes of this study provide a funded base for the evaluation and adaptation of the course content in health education.

## 2. Methods

### 2.1. Study Design and Setting

The observational study was performed between July 2014 and April 2015 by final-year pharmacy interns from Ghent University in 136 community pharmacies across Flanders, Belgium. Since no personal patient data were collected, the compilation of an informed consent was not required. Ethical approval for the study was obtained by Ghent University (B670201731221).

### 2.2. Data Collection

During four months, each pharmacy intern was asked to record the first two food- and nutrition-related cases with which they were confronted every month. A food- and nutrition-related case was defined as “Every question of a patient in the community pharmacy, asked to the pharmacy intern, concerning food, nutrition, or supplements, whether or not linked to the purchase of an over-the-counter (OTC)-product or medicinal drugs, both applicable to humans or animals”. To avoid bias, interns were not allowed to select any cases, so the first cases had to be recorded sequentially. Hence, an independent selection could be achieved. The study was initiated after five weeks of internship (26 weeks in total) to optimize the interns’ acquaintance with their pharmacy work. The pharmacy interns were instructed to treat the cases as a usual care situation.

For each case, the students were asked to fill in a report containing seven questions via an online platform, only accessible via a personalized account: (1) What was the case?; (2) Was the case related to a prescription, an OTC-product, or merely advice?; (3) Whom did the case relate to? (i.e., gender, age, and specific population parameters such as pregnancy, breastfeeding, etc.); (4) Did you have enough knowledge to discuss the case?; (5) Did you search for additional information?; (6) Where did you search this information?; (7) Did you, after looking up extra information, have enough knowledge to discuss the case?

### 2.3. Analysis

To achieve a better insight, two researchers (L.V.H. and E.T.) qualitatively overviewed all the acquired data and classified all cases into 18 main categories, including: interactions between food, food supplements, and drugs; ingredients in food; E-numbers; healthy food and food/nutritional recommendations; vegetarianism and veganism; functional foods and nutraceuticals; food supplements; novel foods and hype-surrounding superfoods; weight-loss diets and products; food allergies and intolerances; high-protein foods and drinks; diet for phenylketonuria; diabetes and diet; enteral and tube feeding; baby food; food for the geriatric population; food for athletes; and food safety (Table 1). Further categorization into specific nutritional topics within these main categories was done post hoc, to enable formulation of recommendations. Cases that were not clear or that were not food-related were excluded from the analysis.

## 3. Results

A total of 1084 cases were recorded by 135 pharmacy interns. One student withdrew from the internship. Eighty cases were excluded after classification. Seventy-eight cases were not considered food-related by the researchers, and two had incomplete information.

### 3.1. Origin of the Cases

Of the 1004 included cases, 46.6% (468/1004) were related to the purchase of an OTC-product, 41.7% (419/1004) considered merely advice, and the remaining 11.7% (117/1004) were related to the purchase of a prescription drug or product.

### 3.2. Case Subjects

The recorded cases applied to diverse case subjects (man, woman, adolescent, animal). Eighteen percent (18%) of cases were related to babies or infants. Four percent (4%) of case subjects involved pregnant women, and 1% involved women with a pregnancy wish. Another 1% of case subjects were related to breastfeeding women. Three percent (3%) of cases involved students. A last category, comprising merely 0.2% of the total cases, considered animals. The remaining 72.8% considered rather general questions and could therefore not be assigned to a specific subject.

### 3.3. Ability of Pharmacy Interns to Discuss Nutritional Cases

In 45% (447/1004) of all cases, pharmacy interns were able to immediately discuss the cases without searching for additional information. These percentages did not increase over the course of the internship, with 42% in the first month, 46% in the second month, 47% in the third month, and 44% in the fourth month. Table 2 shows the percentage of cases in which pharmacy interns had sufficient knowledge to answer before and after research.

Pharmacy interns appeared to have the most trouble with patients that were merely seeking advice. Sixty-two percent (260/490) of advice-related cases could not be answered immediately, against 51% (239/468) of the cases on OTC products. Fifty percent (58/117) of cases on products prescribed by a medical doctor could not be answered. The amount of cases that could not be answered after research was only 5% (46/1004).

### 3.4. Looking Up Additional Information

For 557 cases, the pharmacy interns did not have sufficient knowledge to answer immediately. In 552 of these 557 cases (99%), the pharmacy interns used additional sources of information. Moreover, for 228 out of the 447 (51%) cases, the pharmacy interns reported having sufficient knowledge to answer, but nevertheless consulted additional sources of information before doing so. For the remaining 219 (49%), no additional information was looked up (Table 3).

During the process of looking up additional information, 46% of pharmacy-interns used one single source to find an answer to their question, 39% used two sources, 12% used three sources, and 3% used four sources. The different sources that were used were textbooks, manuals, websites, pharmacy software, farma compendium (i.e., a Belgian specialized notebook for community pharmacies), supervisor’s help (i.e., the pharmacist in the community pharmacy), and an online or telephonic helpdesk. The help from the supervisor and information from websites were manifestly the most used sources. In 31% of cases, the pharmacy interns only consulted their supervisor to obtain the needed information. In 17% of cases, they consulted their supervisor and a website. In 8% of cases, they used only web information.

## 4. Discussion

This observational study analyzed more than 1000 food- and nutrition-related cases addressed in 136 community pharmacies across Flanders, Belgium, during 2014–2015. Cases were most often related to the categories “food supplements”, “baby food”, and “healthy food and food/nutritional recommendations”. Pharmacy interns were able to answer the patients’ food-related questions in about half of the cases. Advice-related questions were the hardest to answer, in comparison to cases related to OTC products or prescribed products. Almost all pharmacy interns looked up additional information when they were unable to answer the question on the basis of prior knowledge. The supervisor’s opinion (i.e., the counselling pharmacist in the community pharmacy) and the Internet were the most frequently consulted sources. After looking up extra information, only a small proportion of cases remained unanswered.

### 4.1. Study Setting in Belgium

In Belgium, health insurance is organized by private, nonprofit sickness funds that are financed by the government. Membership in a sickness fund is mandatory; however, the choice of sickness fund is free [13]. Membership fees range from EUR 0–200 per year. For most ambulatory care and for pharmaceuticals, payments are made mainly through third-payer arrangements, whereby the sickness funds pay the providers and the patients are only responsible for the copayment. Belgium has a very dense network of community pharmacies (one pharmacy for 2300 patients), which have a monopoly on dispensing registered medication. There is no reimbursement for dietician consultations, except for a limited number of visits for diabetes patients. Therefore, the threshold for consulting a dietician is high [14].

### 4.2. How Is Food Science Taught in Health Education?

As for pharmacy education, all universities in Belgium address a limited number of food-related subjects as an elective or major course. The topics that are generally covered are nutrient classes and their nutritional value, food safety, dietary products and nutrition for specific populations, and recommendations for a healthy diet. This is not remarkable. The competency profile of the pharmacist merely mentions food or nutrition. The first mentioning considers the dispensation of food supplements. The pharmacist must know enough about nutrition to frame the use of food supplements. A second point in which nutrition is mentioned is that in terms of prevention, the pharmacist must be able to offer advice about a healthy and balanced diet.

Abroad, food science has a varying order of importance. In Germany (Bonn University and Marburg University) [15,16] and France (Montpellier University and Lille University) [17,18], basic aspects of food science, like nutrient classes, nutritional demands in different life stages, energy balance, etc., are incorporated in a mandatory course. In the Netherlands (Utrecht University and Groningen University) [19,20], the United Kingdom (Greenwich University and University College London) [21,22], and the United States (UNC Eshelman School of Pharmacy, University of Minnesota’s College of Pharmacy) [23,24], merely elective courses or no nutritional courses are present. The Accreditation Council of Pharmacy Education (ACPE), along with the North American Pharmacist Licensure Examination (NAPLEX), however, requires the inclusion of self-care therapy, meaning that OTC-supplementation is likely taught in all curricula in the American context.

When focusing on food supplements, one study showed that students considered their education on dietary supplements to be inadequate [25]. Additionally, multiple studies reported that healthcare professionals had insufficient knowledge about (adverse effects of) dietary and herbal supplements [26,27,28]. However, most studies were specifically targeted toward complementary or alternative medicine, and not to purely nutrition-related patient questions.

### 4.3. What Does This Study Add?

First of all, this study confirmed that food- and nutrition-related cases are often discussed in primary care, and subjects are diverse. The primary care workers in this study, i.e., community pharmacists, could discuss about half of food-related questions without the need for an additional information search. We can deduct that the curriculum had provided a solid knowledge base to discuss these topics. However, for students currently enrolled in a program that does not provide a food-related course, this outcome might be different.

Subsequently, it was a positive finding that almost all interns lacking knowledge to answer a patient’s question consulted additional information sources. Moreover, half of the pharmacy interns that were able to answer the question still searched for additional information. The main sources currently used, however, (the supervisor and the Internet) are prone to bias. The information provided by the supervisor is influenced by his or her opinions and prior knowledge as well. Despite that some students used websites like “BMJ Best Practice” [29], ”UpToDate” [30], or the “Cochrane Library” [31], which are reliable scientific sources, many students used commercial websites or blogs. Course programs must therefore be attentive to including courses on how to search for evidence-based information and how to critically evaluate information.

Finally, this study showed that—besides food supplements—patients also sought information about baby foods, healthy diets, and many other diet-related topics in the community pharmacy.

## 5. Specific Recommendations for Health Education

Reviewing the results of this study, we can make several recommendations regarding health education. First, as food- and nutrition-related topics are numerous and subjects are very diverse, it is impossible to discuss all topics during education. Based on this practice research, we can however recommend a number of core themes and other, optional food- and nutrition-related subjects (Figure 1). We believe the community pharmacist is a trusted healthcare provider worldwide, and these recommendations therefore apply to curricula all over the world.

In addition, every healthcare specialist must be aware of their personal knowledge limitation, and refer to dieticians when they believe the requested information is beyond their scope of expertise. We therefore recommend an interdisciplinary course at the undergraduate level on a medical topic in which the role and competencies of every healthcare worker are visible. Ideal subjects for an interdisciplinary course include diabetes or food intolerances. One might suppose that such a strategy will lead to an improved referral policy (Figure 1). Policy makers should also consider mandatory interdisciplinary courses to be part of postgraduate training and accreditation. This integration is of course dependent on the local situation and the possibility of referral in the respective countries.

Finally, skills to search for evidence-based information is essential. Health education should focus on how to look up reliable scientific information in practice. This will support the adequate answering of patient questions, as well as lifelong learning. The mandatory course introduced in academic year 2020–2021 on philosophy, methodology, and integrity in pharmaceutical research will further develop skills of critical thinking. These recommendations reach much further than only the food-science topics, and represent a skill that is essential to the healthcare provider of the future.

As a specific example, we provide the curriculum change implemented in the Bachelor and Master of Pharmaceutical Sciences program at Ghent University. We decided upon these organizational and content changes based on the result of this study (Figure 2).

### Strengths and Limitations

This is the first practice-based study that provides a broad insight into the most discussed food- and nutrition-related topics in primary care. We were also able to provide a real-life example of a curriculum change in the Bachelor and Master in Pharmaceutical Sciences program at Ghent University based on the study results.

The first limitation in this study concerns the data-collection method performed by different final-year pharmacy interns. The interpretation of the term “food-related case” might differ between pharmacy interns, and therefore, there was a risk of selection bias. The second limitation was the low amount of questions in some categories. Some categories even comprised merely one question. This low amount of questions might interfere with the interpretation of the percentages of questions unanswered. No correct analyses of the interns prior knowledge could be performed from such a limited number. The relevance of some topics might be underestimated, as we only documented cases when patients asked for advice or a specific product. This requires that the patient was aware of a certain problem. For example, in geriatric care, malnutrition is prevalent [32]. However, in this study, malnutrition was merely discussed. This might show that malnutrition is sometimes undetected. In a study in Northern Ireland, for example, qualitative responses implied that pharmacists were unhappy with the way malnutrition was managed in the community, especially concerning compliance and follow-up of intake of oral nutrition supplements [11]. A final limitation was that interns had to self-report on their ability to provide an adequate answer to the questions. This might lead to bias; however, all students were at all times supervised by a registered community pharmacist. We believe that—in case a student would fail to provide a sufficient answer—this would be corrected by the counselor.

## 6. Conclusions

In the frame of an observational study in community pharmacies in Belgium, we researched the topics of food-related questions (n = 1004). Questions/cases about food supplements, baby and infant foods, and nutritional recommendations were numerous. In about half of the cases, pharmacy interns were able to answer the patients’ food-related questions. Advice-related questions were the hardest to answer, in comparison to cases related to OTC products or prescribed products. It was noteworthy that looking up additional information was a prerequisite to answer the defined question on the basis of prior knowledge. The supervisor’s opinion and the Internet were the most frequently consulted sources, highlighting the importance of including qualitative search strategies in the pharmacy’s intern education. Besides this, it remains crucial for all healthcare providers in primary healthcare to have a basic knowledge of food-related topics.

## Figures and Tables

**Figure 1 pharmacy-09-00104-f001:**
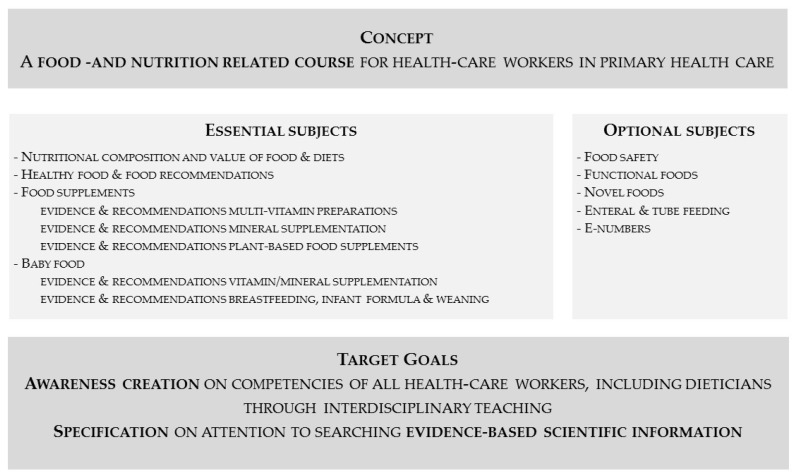
Recommendations for improvements of health education in terms of food and nutrition.

**Figure 2 pharmacy-09-00104-f002:**
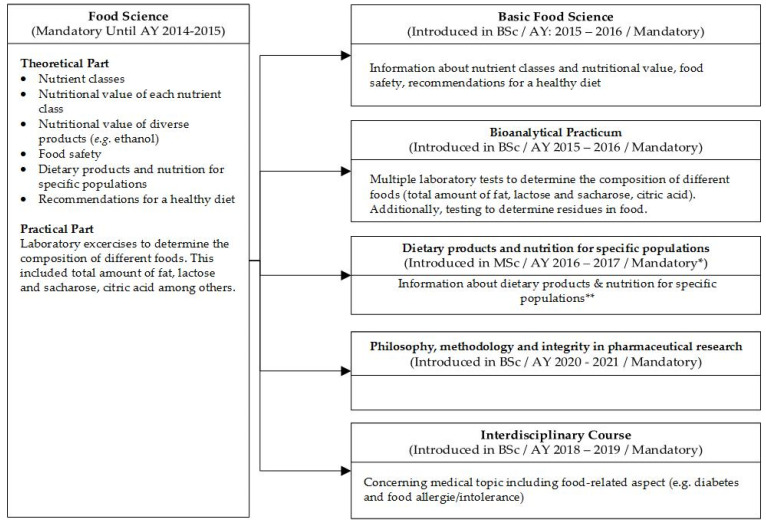
An example of curriculum changes in the Bachelor and Master in Pharmaceutical Sciences program at Ghent University. AY: Academic Year; BSc: Bachelor of Science; MSc: Master of Science. * Mandatory for Master in Pharmaceutical Care, Elective for Master in Drug Development. ** Subjects include (1) legislation, (2) food recommendations in case of food-related diseases (celiac disease, diabetes, gastric bypass, lactose intolerance, etc.), (3) enteral feeding, (4) food for specific (patient) groups (infants and toddlers, athletes, elderly), (5) food allergens, (6) weight-loss products, and (7) food supplements.

**Table 1 pharmacy-09-00104-t001:** Detailed overview of case distribution across 18 nutritional categories (n = 1004).

Category or Nutritional Topic	N°, (%)
**Food Supplements**	**379 (37.8)**
- Multivitamin preparations	52
- Vitamin D, calcium, or combinations	45
- Preconception, pregnancy, or breastfeeding vitamins	34
- Vitamins to improve sleep, diminish fatigue, or give an energy boost	30
- Magnesium	28
- Vitamins for studying	27
- Supplements for specific indications (menopause, arthritis, diarrhea, bladder infection, etc.)	23
- Red yeast rice for hypercholesterolemia or Q10 when statin use	22
- Iron and vitamin B12	15
- Hair and nail vitamins	14
- Omega 3, 6, or 9	13
- Vitamin C	7
- Other	69
**Baby Food**	**187 (18.6)**
- Formula for insatiety	37
- Formula for constipation, diarrhea, colic, or combinations	27
- Switching from formula to follow-on formula or growing-up milk	26
- Formula for reflux and regurgitation	21
- Formula for allergy prevention of treatment	19
- Switching from breastmilk to formula	18
- Switching from milk to solid foods	5
- Formula supplementation when insufficient milk supply	5
- Other (preparation, recommendations, etc.)	29
**Healthy Food and Food/Nutritional Recommendations**	**106 (10.6)**
- Food recommendations in function of disease (constipation, heartburn, gout, diarrhea, migraine, hypertension, hypercholesterolemia, etc.)	59
- Food recommendations in function of vitamin or mineral deficiency (iron, B12, D, folic acid, etc.)	18
- Food recommendations in function of colonoscopy, bowel examination, or gastric bypass	10
- Balanced diet	6
- Food restrictions during pregnancy	5
- Other	8
**Weight Loss Diets and Products**	**78 (7.8)**
- Weight-loss pills or supplements	24
- Meal replacements	20
- Specific diets (low-carb, soup diet, high protein, etc.)	9
- Light products and sugar substitutes	3
- Other	22
**High-Protein Foods/Drinks**	**63 (6.27)**
- Food replacement in illness	29
- Weight gain	12
- Food supplements for the geriatric population	11
- Other	11
**Food for Athletes**	**48 (4.78)**
- Supplements for endurance training	28
- Supplements for building up muscle mass	10
- Other	10
**Interactions between Drugs and Food**	**44 (4.38)**
- Combination of antibiotics and dairy	16
- Combination of medication and alcohol	8
- Combination of food supplements and food	6
- Combination of medication and grapefruit	3
- Other	11
**Food Allergies and Intolerances**	**33 (3.29)**
- Gluten	6
- Lactose	5
- Other (cow’s milk protein allergy, wheat allergy, etc.)	7
- Presence of allergens in drugs	15
**Vegetarianism and Veganism**	**16 (1.59)**
- Vitamin or mineral deficiency	11
- Other	5
**Food Safety**	**11 (1.09)**
**Diabetes and Diet**	**11 (1.09)**
**Food for the Geriatric Population**	**9 (0.90)**
**Ingredients in Food**	**8 (0.80)**
**Functional Foods/Nutraceuticals**	**5 (0.50)**
**Diet for Phenylketonuria**	**3 (0.30)**
**Enteral and Tube Feeding**	**1 (0.10)**
**Novel Foods and Hype-Surrounding Superfoods**	**1 (0.10)**
**E-Numbers**	**1 (0.10)**

**Table 2 pharmacy-09-00104-t002:** Distribution of cases in which pharmacy interns had sufficient knowledge to answer cases before and after research (n = 1004).

Category	Number of Cases (%)	Percentage of Cases in Which the Intern Had Enough Knowledge to Discuss the Case Immediately	Percentage of Cases in Which the Intern Had Enough Knowledge to Discuss the Case after Research (%)
Food supplements	38% (n = 379)	49%	96%
Baby food	19% (n = 187)	47%	97%
Healthy food and food/nutritional recommendations	11% (n = 106)	46%	93%
Weight-loss diets and products	8% (n = 78)	44%	95%
High-protein foods/drinks	6% (n = 63)	47%	95%
Food for athletes	5% (n = 48)	21%	96%
Interactions between drugs and food	4% (n = 44)	54%	93%
Food allergies and intolerances	3% (n = 33)	18%	94%
Vegetarianism and veganism	2% (n = 16)	25%	94%
Food safety	1% (n = 11)	27%	100%
Diabetes and diet	1% (n = 11)	45%	100%
Food for the geriatric population	0.9% (n = 9)	33%	100%
Ingredients in food	0.8% (n = 8)	37%	87%
Functional foods/Nutraceuticals	0.5% (n = 5)	60%	100%
Diet for phenylketonuria	0.3% (n = 3)	0%	100%
Enteral and tube feeding	0.1% (n = 1)	0%	100%
Novel foods and hype-surrounding superfoods	0.1% (n = 1)	0%	100%
E-numbers	0.1% (n = 1)	0%	100%

**Table 3 pharmacy-09-00104-t003:** Distribution of cases on how additional information was sought (n = 1004).

Category	Number of Cases (%)	Category	Number of Cases (%)
No sufficient knowledge to answer immediately	557 (55%)	Used additional sources of information	552/557 (99%)
		Used no additional sources of information	5/557 (1%)
Sufficient knowledge to answer immediately	447 (51%)	Used additional sources of information	228/447 (51%)
		Used no additional sources of information	219/447 (49%)

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
