# Peer review of "Revisiting the Food- and Nutrition-Related Curriculum in Healthcare Education: An Example for Pharmacy Education"

_pharmacy, 2021, doi:10.3390/pharmacy9020104_

Round 1
Reviewer 1 Report
General Comments:
There are a lot of different font sizes utilized in this manuscript (mainly in the abstract). Also, the referencing is not consistent (the period should be before the numbers and not include a space between the period and numbers). These are minor edits that can be easily corrected. There are also minor grammatical errors throughout the manuscript, particularly with regard to commas. Please revise. Overall, the manuscript reads pretty well. The biggest area for improvement is broaden and direct the literature review a bit to be inclusive of what is currently being taught vs areas for expansion. Some areas may be taught more than others. Teaching the how to approach a nutrition discussion as well as how to find information quickly and accurately are key points you start to address, but this could be incorporated into the introduction and conclusion and expanded upon more thoroughly in the discussion.
Abstract:
Results: With 45% immediately discuss, and 5% could not be addressed, are you meaning that 50% could be addressed after looking up extra information? Please make clearer in the abstract…
Introduction
- Paragraph 1, Sentence 2: What obvious impact?
- Paragraph 1: I believe you are trying to say that dietary choices are important, but consumers should turn to valid sources of information vs media. The way this is written sounds very opinionated. While all true statements, think how you could make this flow well and be more factual vs opinion.
- Any accreditation or curricular guidelines that would provide recommendations on what to include in health education?
Methods
- There is good context on the juxtaposition of providers with nutritional consultation. However, what is the training for pharmacists? What are the expectations for pharmacists in terms of providing patient education and counseling?
- Was this a physical report or completed electronically?
- Did each of the researchers classify cases independently and come to consensus on the coding or were they done completely independently?
- What required training do the pharmacy students receive on any form of nutrition? For example, in the US, PharmD students will get training on nutritional supplements and vitamins, but may have varying education on dietary advice. (Some of this is included in your discussion - may want to mention briefly.)
Results
- Consider breaking Table 1 into two tables (instead of continued). It is a bit bulky.
Discussion:
- Overall, I think a more comprehensive literature review needs to be performed and integrated. For example, I think the literature review on inclusion in curricula internationally is a bit limited.
- US Food Science: As a clarification, ACPE requires the inclusion of self-care pharmacotherapy. These concepts are also included on the NAPLEX. Thus, OTC supplements (vitamins, minerals) are likely taught in all curricula; other food-related elements will vary.
- Some of these articles may be helpful: (None are self-citations)
- Axon DR, Vanova J, Edel C, Slack M. Dietary Supplement Use, Knowledge, and Perceptions Among Student Pharmacists. Am J Pharm Educ. 2017;81(5):92. doi:10.5688/ajpe81592
- Scaletta A, et al. Complementary and alternative medicine education in U.S. schools and colleges of pharmacy. Curr Pharm Teach Learn 2017;9(4):521-527. https://doi.org/10.1016/j.cptl.2017.03.009
- Pokladnikova J, Lie D. Comparison of attitudes, beliefs, and resource-seeking behavior for CAM among first- and third-year Czech pharmacy students. Am J Pharm Educ. 2008;72(2):24. doi:10.5688/aj720224
- James, P.B., Bah, A.J. Awareness, use, attitude and perceived need for Complementary and Alternative Medicine (CAM) education among undergraduate pharmacy students in Sierra Leone: a descriptive cross-sectional survey. BMC Complement Altern Med 14, 438 (2014). https://doi.org/10.1186/1472-6882-14-438
- Consider in your recommendations section to integrate peer-reviewed literature to support key items to include as well as drug information skills. There should be quite a few, at least on drug information. You may also consider creating a key strategy on approaches to obtaining information needed during an inquiry from the patient (collecting information) in order to provide an appropriate response.
Conclusion:
- Consider adding in information about their needing to look up information or consult with the pharmacist. That was a key point.
Author Response
We want to thank the reviewers for their valuable comments and remarks. We have highlighted the revisions via track changes in the manuscript, along with a clean version, and we have inserted a more detailed rebuttal in the attached pdf document

Reviewer 2 Report
Introduction:
Would suggest the Aim be looked at again to more fully incorporate the extent of the data presented. e.g. include about the sources gone to increase knowledge and to support the answering of questions.
Methods:
Would suggest some text be moved and better suited to discussion e.g. aspects about the Belgium Health Insurance.
Greater clarity around the case subjects would be useful
Why were the 18 categories pre-defined instead of using the data qualitatively to develop the themes/categories.
Results:
Table 2 Relook at the presentation. for consistency perhaps think about formatting as % (n=) throughout
Additional information section This may be better presented in tabular form including numbers for how often the different sources were used. This has obviously been considered when you read further in the Discussion
Discussion:
Figure 1 Be clear in the recommendations as to whether this is suggested for Belgium / Europe/ Globally
Some additional documentation about inclusion of dietitians. It feels as if they are singled out and this can be perceived in different ways. It reads that it is in context of interdisciplinary teaching. Is this correct? Is this at undergraduate level or as practicing health care professionals? It also says it will lead to an improved referral policy. Query to whom? Some clarity and reformatting of the Figure would be beneficial.
Good strengths and weaknesses identified.
Conclusion:
Could be added to in order to fully incorporate the work presented.
Author Response
We want to thank the reviewers for their valuable comments and remarks. We have highlighted the revisions via track changes in the manuscript, along with a clean version, and we have inserted a more detailed rebuttal in the attached pdf document.

Round 2
Reviewer 1 Report
Thank you for your edits. The key area to correct is in the discussion:
- As a clarification, ACPE requires the inclusion of self-care pharmacotherapy. These concepts are also included on the NAPLEX. Thus, OTC supplements (vitamins, minerals) are likely taught in all curricula; other food-related elements will vary. It is not correct to state that we only include it in elective courses.
Otherwise, no further comments.